

# In field use of water samples for genomic surveillance of infectious spleen and kidney necrosis virus (ISKNV) infecting tilapia fish in Lake Volta, Ghana

Shayma Alathari[1], Andrew Joseph[2], Luis M. Bolaños[1], David J. Studholme[1], Aaron R. Jeffries[1], Patrick Appenteng[3], Kwaku A. Duodu[3], Eric B. Sawyerr[3], Richard Paley[2], Charles R. Tyler[1,4] and Ben Temperton[1]

[1] Faculty of Health and Life Sciences, University of Exeter, Exeter, United Kingdom
[2] Centre for Environment, Fisheries and Aquaculture Science (Cefas), Weymouth, United Kingdom
[3] Fisheries Commission, Ministry of Fisheries and Aquaculture Development, Accra, Ghana
[4] University of Exeter, Sustainable Aquaculture Futures Centre, Exeter, United Kingdom

## ABSTRACT

Viral outbreaks are a constant threat to aquaculture, limiting production for better global food security. A lack of diagnostic testing and monitoring in resource-limited areas hinders the capacity to respond rapidly to disease outbreaks and to prevent viral pathogens becoming endemic in fisheries productive waters. Recent developments in diagnostic testing for emerging viruses, however, offers a solution for rapid *in situ* monitoring of viral outbreaks. Genomic epidemiology has furthermore proven highly effective in detecting viral mutations involved in pathogenesis and assisting in resolving chains of transmission. Here, we demonstrate the application of an in-field epidemiological tool kit to track viral outbreaks in aquaculture on farms with reduced access to diagnostic labs, and with non-destructive sampling. Inspired by the "lab in a suitcase" approach used for genomic surveillance of human viral pathogens and wastewater monitoring of COVID19, we evaluated the feasibility of real-time genome sequencing surveillance of the fish pathogen, Infectious spleen and kidney necrosis virus (ISKNV) in Lake Volta. Viral fractions from water samples collected from cages holding Nile tilapia (*Oreochromis niloticus*) with suspected ongoing ISKNV infections were concentrated and used as a template for whole genome sequencing, using a previously developed tiled PCR method for ISKNV. Mutations in ISKNV in samples collected from the water surrounding the cages matched those collected from infected caged fish, illustrating that water samples can be used for detecting predominant ISKNV variants in an ongoing outbreak. This approach allows for the detection of ISKNV and tracking of the dynamics of variant frequencies, and may thus assist in guiding control measures for the rapid isolation and quarantine of infected farms and facilities.

Corresponding authors
Shayma Alathari, sa655@exeter.ac.uk
Ben Temperton,
b.temperton@exeter.ac.uk

## INTRODUCTION

Today, 811 million people globally suffer from hunger and 3 billion cannot afford a healthy diet. The United Nations has listed Zero Hunger as one of the global sustainable development goals and to end extreme poverty by 2030 (*Boykin et al., 2018*). As populations continue to grow, aquaculture is expected to play an increasingly important role in improving food security, and most notably in low- and middle-income countries (*Cai, 2022*). New strategies have been developed such as "Blue Transformation" to enhance the role of aquaculture in food production, by providing the legal, policy and technical frameworks required to sustain growth and innovation systems to do so (*FAO, 2022*). Despite significant increases in aquaculture output in the last few decades, all forms of aquaculture are limited by infectious diseases (*FAO, 2019*). Fish disease is usually triggered by poor water and poor farm management and inadequate biosecurity practices (*Ragasa et al., 2022*). Implementation of biosecurity measures in resource-limited countries is, in part, challenging due to a lack of suitable real-time and/or effective diagnostics.

In Ghana, ISKNV, a Megalocytivirus, has become endemic in tilapia in Lake Volta, following a series of outbreaks in 2018 and this has significantly affected local farmers and their livelihoods (*Ramírez-Paredes et al., 2021*). Lake Volta is experiencing ongoing disease, as new cohorts of juvenile fish enter the system. There has been no effective control across multiple farms in what is essentially one large epidemiological unit. According to these farmers, attempts to minimise the impact of outbreaks through heat shocking fish, to reduce the virulence of the virus, or increasing fingerling production, have not helped to improve total production. Genome sequencing provides an unparalleled ability to track infectious disease outbreaks, from the initial detection to understanding factors that contribute to the geographical spread. Indeed, it is emerging as a critical tool in real-time responses to these outbreaks, by providing insights into how viruses transmit, spread and evolve (*Gardy, Loman & Rambaut, 2015*; *Quick et al., 2017*). Accurate reconstruction of strain-resolved genomes is useful to monitor the outbreak of viruses, to track their evolutionary history and develop effective vaccines and drugs, as well as detect the emergence of novel variants that may impact the course of an epidemic (*Luo, Kang & Schönhuth, 2022*; *Child et al., 2023*).

In aquaculture, monitoring large numbers of infections through tissue sampling poses challenges in large-scale outbreaks, particularly in resource-limited settings, as it is time consuming and requires well practised personnel. In human health, analyses of wastewater samples have been used to understand mutations and infection dynamics, as well as an early indicator of infection (*Dharmadhikari et al., 2022*). This method was used to monitor the ongoing evolution of SARS-CoV-2 during the pandemic, and the water-based epidemiological programmes has provided insights into its prevalence and diversity in different communities and detecting the emergence and spread of variants (*Brunner et al., 2023*). In the context of fish pathogens, water-based epidemiology provides a non-invasive routine method to early detection of viruses in asymptomatic fish and ongoing infections, reducing the sacrifice of fish for testing.

In this study, we tested the utility of an in-field water sampling method for whole genome sequencing of ISKNV, using a tiled PCR method that we developed previously (*Alathari et al., 2023a*), as a potential alternative to destructive tissue sampling for genomic surveillance of a disease outbreak in Lake Volta, Ghana. We show water samples collected in the immediate vicinity of the cage fish showed similar variants to infected tissue samples in tilapia at that site, providing confidence in-field water sampling method for genomic surveillance.

## MATERIALS & METHODS

### Samples

In an ongoing outbreak of ISKNV, water and tilapia tissue samples were collected from six geographically distinct Nile tilapia farms (*Oreochromis niloticus)* situated on Lake Volta, Ghana, in January 2023, (see Fig. 1 & Table 1), with approval granted by the Fisheries Commission, Ministry of Fisheries and Aquaculture Development, Accra, Ghana (FC/10/V11/65). Water samples (250–500 mL) were collected from high density cage-based farms (average 2000 fish per 5 m$^3$ cage) on the lake and processed by sequential filtration through a 0.45 μm pore (PES filters), 0.22 μm pore (Durapore PVDF Membrane; Merck, Millipore), and finally concentrating viral particles on 0.1 μm pore filters (Merck, Millipore (Durapore PVDF Membrane)), housed within Luer-lock syringe-compatible casings. An Erwin® quick-grip minibar clamp (6") was used to facilitate the pumping of the water, with a custom 3D-printed adaptor for the syringe (Fig. S1). Portions of this text were previously published as part of a preprint (*Alathari et al., 2023b*).

Viruses on 0.1 μm filters were preserved *in situ* by addition of RNALater®, filling the filter housing, and the inlet and outlet of the filters were sealed with Parafilm®. Filters were transferred to the University of Exeter for further processing. For matching tissue samples, a total of 12 fish were selected from each of the six farms, typically four fish from each of three cages across various fish life stages. Fish were humanely euthanized with a lethal overdose of tricaine methanesulfonate 1,000 mg/g (Pharmaq, Hampshire, UK), and the spleen, liver and kidney were collected on site. Tissue samples were either processed in the field (*Alathari et al., 2023a*), or were preserved in RNALater®, and taken for further processing at the University of Exeter. Fish size, life stages, and any observed clinical signs are detailed in Table S1. For the samples from farm (F), one cage (number three) had been heat-shocked by the farmers as part of their routine treatment before sampling (timeframe unknown).

### DNA extraction

DNA extraction from viral filters was undertaken using the Total nucleic acid Extraction Kit (MasterPure complete DNA/RNA purification kit; Epicenter). Using a Luer-lock syringe (shown in Fig. 2) RNALater® liquid was flushed from the filter's housing prior to adding the extraction buffer. Extraction buffer was prepared by adding 2 μL from the supplied Proteinase K to one mL of the either X1 T+C lysis solution or Red Lysis buffer, resulting in 100 μg mL-1 Proteinase K concentration. A total of one mL of the extraction buffer was gently pushed from the outlet to the inlet of the filter using a three mL syringe. A
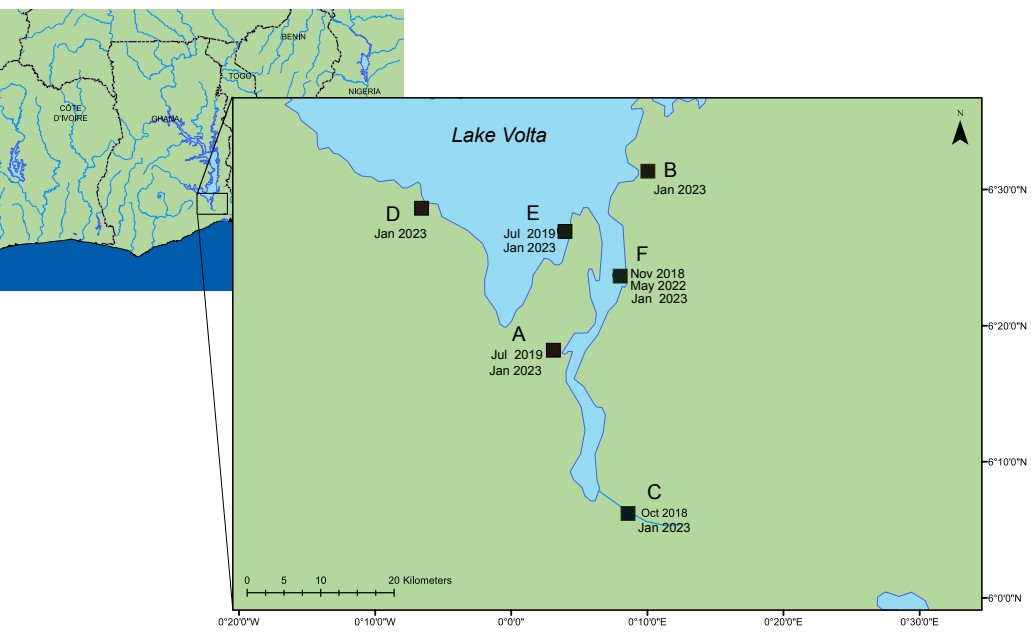

**Figure 1** **A map of the lower region of Lake Volta.** Farms sampled between 2018–2023. Date of sampling and sample ID are listed in Tables S1 & S2. This map was constructed using ArcGIS (GIS software). Version 10.0. Redlands, CA, USA: Environmental Systems Research Institute, Inc., 2010.

**Table 1** **Labelling system for fish farms on Lake Volta, and a comparison with labels in previous study (*Ramírez-Paredes et al., 2021*).**

| Farm name (current study) | Location | Farm name (Previous study) |
|---|---|---|
| *A* | Akosombo | Farm 3 (near farm 7) |
| *B* | Dodi | New |
| *C* | Akuse | Farm 1 |
| *D* | Akaten | New |
| *E* | Dasasi | Farm 6 |
| *F* | Asikuma | Farm 2 |

further three mL syringe was connected to the filter inlet and the assembly was placed into a rotating incubator for 15 min at 65 °C in a hybridization oven (*Steward & Culley, 2010*; *Mueller, Culley & Steward, 2014*). The assembly was removed and allowed to cool briefly at room temperature. The extract was pulled into the aspiration syringe and transferred into a two mL microcentrifuge tube and chilled on ice for 3 min. One-half volume of MPC protein precipitation reagent was added and vortexed for 10 s. The debris was pelleted by centrifugation at 20,000 × g for 15 min at 4 °C, and the supernatant was transferred to a sterile two mL microcentrifuge tube, adding 1 $\mu$L of polyacryl carrier to the sample. An equal volume 100% isopropanol was added and mixed by inverting the tube. The sample was centrifuged at 20,000 × g, for 45 min. The supernatant was then discarded, retaining the pellet, which was washed twice with one mL of 70% ethanol and centrifuged for 1 min.

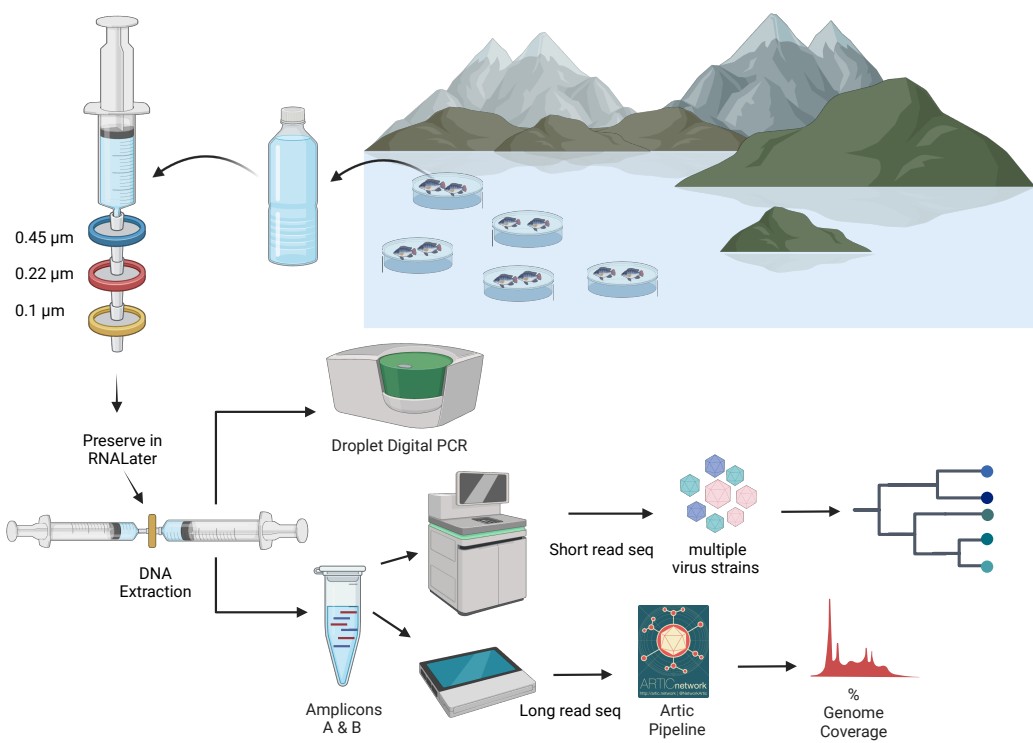

**Figure 2** **An overview of processing of water samples from around the tilapia cages on Lake Volta.**
The figure illustrates the concentrating of ISKNV onto filters, through to DNA extraction, quantification, and sequencing for variant detection. Figure was generated with BioRender (https://biorender.com/). Artic pipeline logo taken from https://artic.network/ncov-2019/ncov2019-bioinformatics-sop.html.

The pellet was air-dried, then dissolved in a 35 $\mu$L elution buffer (EB, NEB) heated to 50 °C. An additional water sample from farm (F) was eluted in nuclease free water (NFW, Ambion).

DNA extraction from tissue samples was performed using the DNeasy Blood and Tissue kit (Qiagen, Manchester, UK), with a starting material of ~10 mg of tissue from pooled organs (liver, kidney and spleen), which were dried for 5 min prior to DNA extraction using the manufacturer's protocol. The nucleic acid, eluted in Elution Buffer, was stored at 4 °C until processing. Quantification of DNA for water samples was performed using the high sensitivity reagents for the Qubit Fluorometer, with broad range reagents used for the tissue samples. Tissue samples were given an alphanumeric name in the format <farm>.<cage>.<fish>.

A positive control for water samples was used to test the efficiency of the DNA extraction method. This was done by spiking water with ISKNV viral particles collected from the 2019 outbreak from Lake Volta (lot: PM 38259) and passaged on BF-2 and or GF cell lines at Cefas. Infected cell lines were stored at −20 °C and thawed at room temperature. A total volume of 40 mL of water was spiked with 0.4 mL of viral particles at a titre of $10^4$–$10^5$ copies/mL. Generally, cell debris was removed by centrifugation at 900 × g for 20 min, and the clarified supernatant was retained to recover the virus for DNA extraction.

Isolated virus from supernatant was filtered and DNA extracted, as mentioned above for field samples.

## Droplet digital PCR for viral quantification

To quantify the number of template strands of ISKNV in water samples, a droplet-digital PCR (ddPCR) amplification test was performed, using an Evagreen assay, described in (*Alathari et al., 2023a*), in accordance with the manufacturer's instructions (Bio-Rad, Hercules, CA, USA). The positive control mentioned above was used as a positive control for viral quantification and detection using the ddPCR. The concentration of DNA input and results are shown in Table S2.

For tissue samples, a probe-based ddPCR assay, using primers and probes by *Lin et al. (2017)*, were used following the manufacturer's instructions (Bio-Rad), generating a 22 µL reaction. This was achieved following the same method described for the Evagreen assay, except the total concentration of the forward and reverse primer was 900 nM, and a concentration of 200 nM for the probe. The DNA volume template added was different according to sample concentration (Table S2).

## Tiled PCR

Extracted DNA from filtered water and tissue samples was quantified using a Qubit fluorometer, and a tiled PCR approach was performed to generate 2kb amplicons for sequencing. For water samples a total of 5µL of each DNA template was added to the reaction, and 1µL of DNA was added for the tissue samples (concentrations are listed in Table S2). The amount of DNA template in the water sample from farm (D) was too high and failed to amplify, therefore the amount was reduced to 2.5 µL. For tissue samples, 0.1 µL of extracted DNA was taken forward for the tiled PCR (Table S2). Two primer pools were prepared with alternating primer sets, described in (*Alathari et al., 2023a*), and Q5 Hotstart High-Fidelity Polymerase (NEB) was used for amplification. Amplicons were quantified using the Qubit dsDNA BR kit (Invitrogen, Waltham, MA, USA), and the two pools (A & B) of amplicons were combined.

## Library preparation and sequencing
### Long read sequencing

*a. Water samples:.* Amplicons generated from water samples from each farm and the prepared mock sample were taken forward for sequencing. Library preparation was performed using the Ligation Sequencing kit 1D (SQK-LSK109) (ONT) and Native Barcoding system (EXP-NBD104) (ONT), according to the manufacturer's instructions, and following the Native barcoding amplicon protocol: version *NBA_9093_v109_revD_12Nov2019*. Equimolar amounts of each barcoded sample were pooled and taken forward for the adaptor ligation step using a total volume of 60 µL of DNA, 5µL of Adaptor Mix II (AMII), and 25 µL of Ligation Buffer (LNB) and 10 µL of T4 DNA Ligase were all added to the barcoded DNA. The reaction was incubated for 10 min at room temperature, and a $0.5 \times$ AMPure XP bead clean-up was performed, followed by $2 \times 250$ µL of Short fragment buffer (SFB) washes. The pellet was then resuspended in 15 µL of Elution Buffer (EB) for 10 min at 37 °C. 15 µL of the elute was retained and ~1 µg

of adaptor ligated DNA was taken forward for priming and loading onto a FLO-MIN 106 (R9.4.1) flow cell.

A MinION run was performed for ~70 h, and the flow cell was refuelled with Flush buffer (FB) after 25 hrs from the start of the sequencing run. All generated sequences were basecalled using the Oxford Nanopore Guppy tool, version v.6.0.4 with super high accuracy, and demultiplexed using guppy_barcoder Reads were trimmed at 1800-2200 bp. Downstream analysis was performed using the Artic Network pipeline to produce a consensus sequence using nanopolish, and the percentage of genome recovery with at least $20 \times$ coverage was calculated (*Alathari et al., 2023a*). All sequences were visualised and polymorphisms were evaluated in Geneious Prime 2022.1.1.

*b. Tissue samples matching water samples.* ONT updated their flow cells during this study, therefore a second library was prepared using the new R10.4 flow cell, to evaluate impact on variant calling. One tissue sample was selected from the same water sampled cages. One filter sample from farm F and one positive control filter sample (both previously sequenced), were sequenced alongside the matching tissue samples from the same cage, as a positive control, and were barcoded using the Native Barcoding kit SQK-NBD114-24. Real-time basecalling was performed on MinKNOW version 23.04.5 with super high accuracy, to produce pod5 files, and demultiplexed with a requirement for barcodes on both ends and a minimum average q-score of 10. The total run was for ~22 hrs. Pod5 files were converted to fast5 files and downstream analysis was performed in a similar way to all previous samples except using Medaka (v.1.4.3) was used instead of nanopolish for variant calling, due to incompatibility between nanopolish and R10 data. Reads were processed using the Artic MinION method of the Artic bioinformatics pipeline: (https://artic.network/ncov-2019/ncov2019-bioinformatics-sop.html).

*c. All tissue samples.* All amplicons generated from tissue samples that produced a visible band on gel electrophoresis following the tiled PCR, and where quantification indicated a concentration more than 10 ng/µL, were taken forward for sequencing. Samples that showed less than 400 viral templates/µL in a ddPCR assay were not taken forward for sequencing (*Alathari et al., 2023a*). A total of 259 ng of DNA was loaded to a FLO-MIN 106 (R9.4.1) flow cell with following library preparation using the Ligation Sequencing kit 1D (SQK-LSK109) (ONT) and Barcoding system (EXP-NBD104) (ONT), according to the manufacturer's instructions: version *NBA_9093_v109_revD_12Nov2019*. The total run was for 72 hrs. A total of 5.24 million reads were generated, and the reads were processed as described above.

### Short read sequencing

In contrast to tissue samples, where a fish is assumed to be infected by a single variant of ISKNV, water samples capture the population of variants circulating within a population. In such samples, consensus basecalling to remove read error from ONT reads is unable to discriminate between natural variation and sequencing error. Therefore, water samples with at least 100 copies of ISKNV/µL were selected to be sequenced (farms C, D, F) using short read sequencing to identify the variants circulating the floating cages in the lake and

determine if more than one variant was present. DNA was extracted as previously described and a tiled PCR was performed using the v2 primers (*Alathari et al., 2023a*), to generate 2 kb amplicons spanning the full genome, followed by 0.6 × bead clean-up with AMPure XP beads. Library preparation was performed with the DNA NEB PCR-free kit, followed by sequencing using the Illumina NovaSeq 6000 using a SP 300 flowcell. Short read sequences were trimmed using Artic guppylex, and mapped against the ISKNV reference genome from the NCBI (NC_003494) with minimap2 (*Li, 2018*) to generate a bam file, which was visualised in Geneious (*v.* 2022.1.1). Reads were visualised and polymorphisms were identified in Geneious and IGV (*v.* 2.16.2).

### Phylogeographic analysis

A phylogeographic tree was constructed comprising 52 whole genome sequences from fish samples collected between 2018–2023, from *Alathari et al. (2023a)*, and this study (Table S3). Consensus genomes were aligned using the *augur* toolkit version 3.0.6 (https://github.com/nextstrain/augur) in Nextstrain, where sequences were aligned using MAFFT (*Katoh et al., 2002*), and a phylogeny was reconstructed using IQ-Tree (*Nguyen et al., 2015*). The tree was further processed using augur translate and augur clade to assign clades to nodes and to integrate phylogenetic analysis with the metadata, where finally augur output was exported and visualised in auspice (https://github.com/nextstrain/auspice) (*Hadfield et al., 2018*). All the consensus sequences generated from each sample were aligned to the ISKNV reference genome, accession no. (NC_003494).

## RESULTS

### ISKNV detection and quantification in tissue and water samples

DNA extraction was performed for tissue samples collected from six different locations across Lake Volta, with matching water samples taken at five locations. Quantification of DNA for all samples was performed using the Qubit Fluorometer and are provided in Table S2.

ddPCR was used to detect and quantify the number of template strands of ISKNV in the extracted DNA from each tissue and water sample. Tissue samples were dominated by non-ISKNV DNA (most likely host DNA). There was an average of 317.45 ng/µL of DNA for all tissue samples, however ddPCR revealed low ISKNV viral template copies in most tissue samples; 71% of the tissue samples had fewer than 100 copies/µL, and in 14 out of the 74 tissue samples no ISKNV was detected, mainly in fish sampled from farms (B) and (C). For water samples, the highest DNA concentration seen, at 21.4 ng/µL, was collected from farm (C).

The number of ISKNV templates in samples collected from water and tissue samples varied considerably across the different farm sites (Fig. 3). At farm (C), tissue samples contained on average only five copies/µL, while the matching water sample had 174 copies/µL (Fig. S2). The average concentration of ISKNV templates found in tissue samples collected from farm (D) in contrast was much higher at 70.6 copies/µL except for one sample (D.3.3) and with very high viral templates, at 382,700 copies/µL, from one fish fingerling. The water samples collected from this cage site also had a high concentration

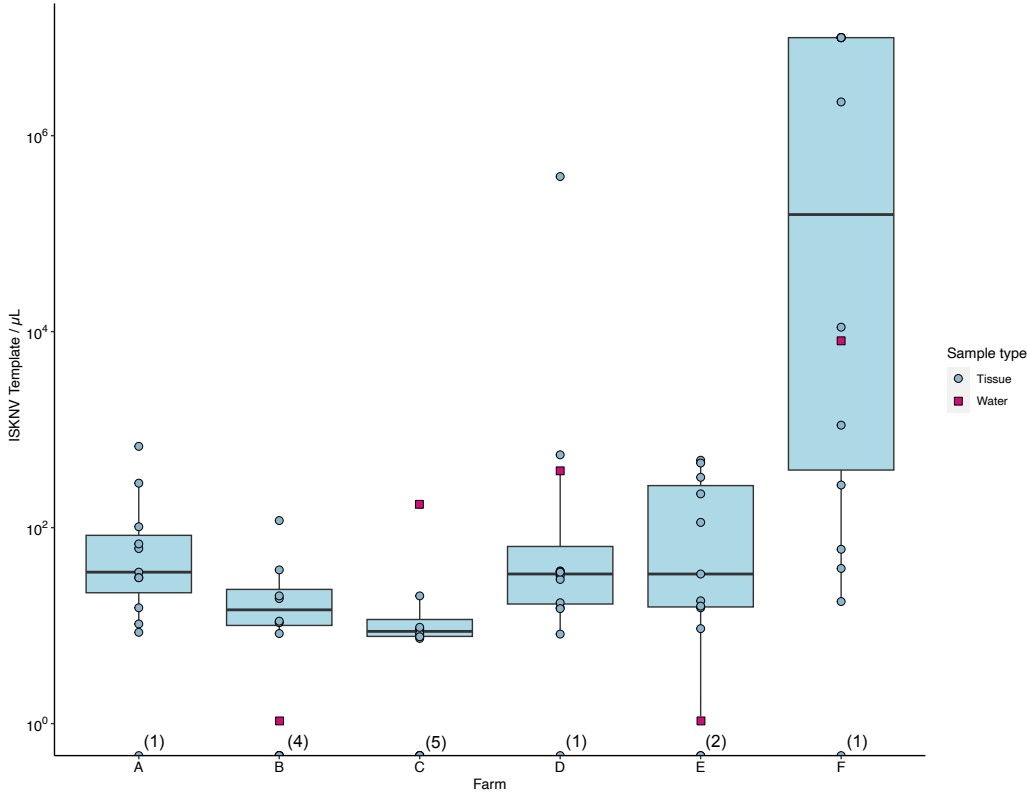

**Figure 3** **The number of viral templates of ISKNV in tissue and water samples collected from the ISKNV outbreak of 2023 in Lake Volta, Ghana.** Distribution of ISKNV template strands in tissue samples (blue) and water samples (red). The number of samples with no ISKNV detected are given parentheses on the *x* axis.

of ISKNV at 361.2 copies/µL. The highest concentration of ISKNV in water samples was seen at farm(F), at 7,560 ISKNV copies /µL, followed by farms (D) & (C), respectively. Contrasting with these farms, (B) and (E) had very low concentrations of ISKNV in the water (~1 copy/µL). Despite the low water concentration of ISKNV at farm E tissues samples had a high ISKNV copy number, with at least 200 copies/µL. In six tissue samples collected from farm (F), the ddPCR failed to provide an accurate count. This was due to saturation of positive droplets at high concentration of DNA template, and this persisted despite further testing with a 20-fold dilution. Negative samples showed no viral template, while the mock filter sample (using viral particles harvested from cell culture) contained 1,584 ISKNV copies/µL. Heat-shocked fish samples from one cage in farm (F) showed no difference in the concentration of ISKNV compared with untreated(non-heat shocked) fish.

Spatial distribution of ISKNV detected across Lake Volta, showed the two farms (B, E) with very low concentrations of ISKNV in the water were both floating cages located far away from other farm cages, and were furthest from the shore (approximately 12 km). The highest titre of ISKNV, were seen in water samples collected from farm (F), and the highest

concentration of ISKNV in tilapia were in juveniles and fingerlings. Moreover, fish in this farm showed the most obvious clinical signs and were experiencing ongoing mortality (Table S1). In general, all life stages were positive for ISKNV, but the lowest concentrations were seen in adult fish.

A tiled PCR was performed on each sample, followed by a gel electrophoresis for each pool. All water samples yielded bands at 2kb, indicative of amplification of ISKNV. Bands for farms (B) & (E) were faint, supporting low template concentrations as measured by ddPCR (Fig. S3). Farm (E) showed multiple bands, with the strongest bands at 1kb. Despite some samples showing faint bands, all the tiled PCR products with any bands at 2kb were taken forward for sequencing.

## Sequencing and phylogeographic analysis for all samples collected from Ghana—Changes to MinION chemistry do not affect our tiled PCR method

A total number of 4.93 M reads were produced from the five water samples, and a total of 5.23 M reads were generated from the five matched tissue samples. The final sequencing run for ISKNV collected from tissue samples was 1.19 M reads. The median length of all samples is reported in Table S3.

When compared to the ISKNV reference genome, the greatest proportion of the whole genome recovered was 98.18% in a tissue sample of a fingerling from cage 4 at farm (F). The highest genome recovery for water samples was 97.49%, collected from the same cage at farm F. Additionally, one sample (from fingerling tissue) from farm (D) had high genome recovery of 97.51%, matching water samples that showed high concentration of ISKNV by ddPCR, and sequencing resulted in genome recovery of 85.6%. Around two-thirds of all sequenced samples recovered at least 50% of the full ISKNV genome. In our previous study, we identified a minimum requirement of 482 copies/µL of ISKNV to yield a genome with >50% recovery (Alathari et al., 2023a). Here, in water samples with fewer than 482 copies/µL of ISKNV produced more than 50% of genome recovery, suggesting lower input requirements for water samples due to an unknown mechanism. A list of genome recovery for each sample is provided in Table S3.

Phylogeographic analysis was performed to investigate the epidemiology of ISKNV virus and disease in Lake Volta, and as a potential indicator of transmission for which closely related genomes indicate closely related infections, shown in Fig. 4. For all except one case, the tissue samples collected from farms (E) and (F) in 2023, formed a separate clade, including the two water samples collected from those farms, and the water sample from farm (C). The 2023 tissue sample from farm (C) along with the tissue and water samples from farm (D) grouped together closely though were separate to earlier samples from the same farm (2018–2022). The highest divergence was seen in samples collected in 2023 from farm F sample (F.3.2), and was related most closely to samples collected from the same farm in 2022.

A group of samples collected from (F) in 2022 diverged from a clade of samples from a previous sampling at this location, clustering separately, due to a mutation occurring
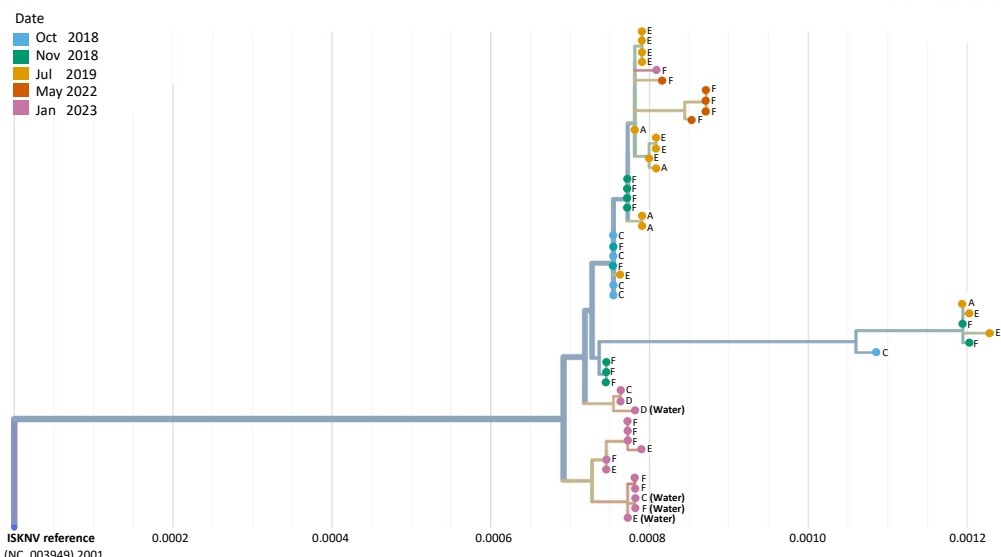

**Figure 4** **A phylogenetic tree of full ISKNV genomes from samples collected from Lake Volta, Ghana since 2018.** Tissue and water samples collected from the latest outbreak were included and the colour represents the date of sampling. The tree was produced in Nextstrain (*Hadfield et al., 2018*).

in the major capsid protein(MCP) that is unique to these samples. Genome recovery and variant detection was comparable between R9 and R10 flow cells.

To investigate differences of mutation profiles between genes across the ISKNV genome, we compared the percentage of polymorphic positions in any ORF for each of the genomes sequenced using the original ISKNV genome as a reference (Fig. 5). The genomes selected were those that had 80% of genome recovery or above, compared with the reference ISKNV genome (2001), with remaining genomes removed from the analysis to avoid spurious single nucleotide polymorphisms (SNPs) from low coverage. Additionally, the repeat region (ORF025) was removed, as this represents a gene duplication and a potential region for circular permutation of the genome, rather than a coding region.

The highest percentage of mutations per gene were in ORF074 and ORF059, which have no assigned function. In general, ISKNV samples collected from Ghana had similar mutations, but samples collected from 2023 had mutations in samples collected from farms (E) and (F) which were not observed in any samples collected throughout previous years samplings. Mutations in the ORF004 were exclusively seen in four samples collected from 2018 with an outlier sample from (F.3.2) collected in 2023, which may explain its divergence from the clade of the outbreak of 2018 on the phylogenetic tree (Fig. 4). All samples collected from Ghana shared a mutation in the ankyrin repeat protein (ORF125), an immunogenic gene, while another immunogenic gene (ORF117) showed a mutation only in a sample collected from farm (D). This mutation was also seen in the matching water sample. All samples had a mutation in ORF022, a proposed virulence gene, except one sample collected from 2018. Mutations in the MCP (ORF006) were higher in samples

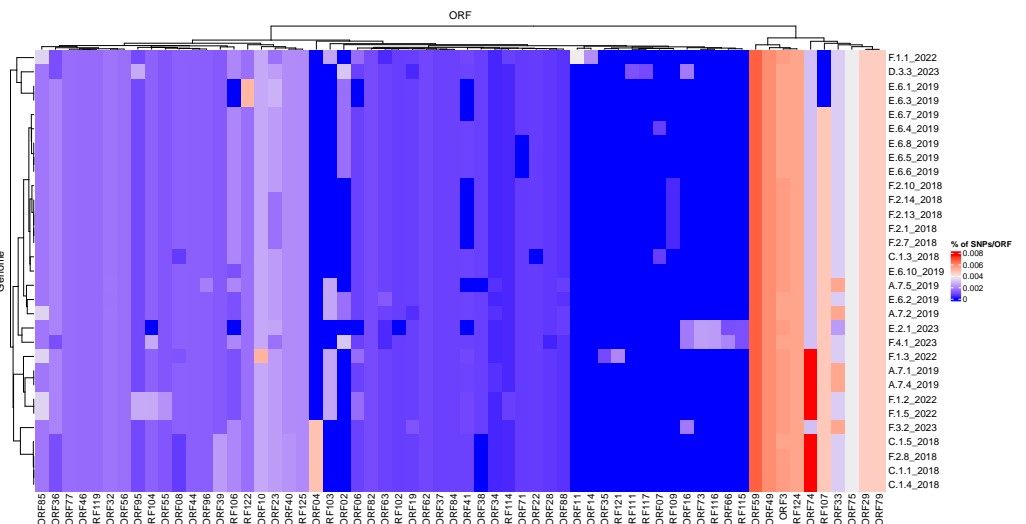

**Figure 5 Mutational frequencies within the ISKNV genomes of fish tissue samples in Lake Volta, Ghana, since 2018.** Heatmap shows the percentage of mutations per gene (ORF), represented on the *x* axis. Genomes with less than 80% genome recovery and ORFs with no mutations were removed, as well as ORF025 (repeat gene).

collected in 2022 than all other samples due to two mutations at this location for four out of five samples.

Short read sequences of the water sample from farm (F), produced a total of 7,915,456 reads. Manual curation of the data using IGV (*v.* 2.16.2) showed the number of identified SNPs to be different to the number of SNPs detected using Geneious when using the default parameters to annotate and predict SNPs. A total of 86 SNPs were observed in IGV, while only 58 SNPs (46 SNPs with 200 × coverage) were listed in Geneious, with five deletions, and two insertions. A total of 27 of these were non-synonymous mutations. In comparison, the consensus sequence for the same sample generated using long read sequencing showed 46 SNPs with two insertions and four deletions. When examining the alignment in Geneious and variant/SNP calling using annotation default settings, some locations, such as a SNP in ORF058- C51,475T (coverage 4,282) was found to have a variant frequency of 90.7%, where 8.1% belonged to the original reference sequence. A similar SNP was manually detected in ORF040, location (C40,742T); however, this SNP was not detected by the Geneious software using the "annotate and predict" feature.

Long read sequencing of water sample from farm (F) showed 33 SNPs in common with short read sequences, and the same mutation at ORF058 with variant frequency of 89%, where only two fish tissue samples collected from the same farm showed the same mutation. Long read sequences from both water and tissue samples from farm (F), had 51 SNPs in common, with three extra SNPs that were unique to the water sample, and another three unique to tissue samples. Polymorphisms and substitutions were annotated in Geneious for short read and long read sequences from farm(F) listed in Table S4.

In addition, water samples from farms (C) and (D) were sequenced using short read sequencing, and produced 21,394,234 and 16,788,272 reads, respectively. Annotation in Geneious detected 34 non-synonymous mutations out of a total of 48 SNPs in samples from farm (D), where 33 SNPs had at least 200x coverage, and four deletions. The unique SNP for ORF0117 in water and tissue samples collected from this farm using long read sequencing, was confirmed in short read sequencing. On the other hand, annotation detected 33 non-synonymous mutations out of a total of 48 SNPs in the farm (C) sample, 39 of these SNPs had at least 200x coverage. The two SNPs mentioned above in farm (F) were detected in short reads from farm (C) but not in the farm (D). Finally, a mutation in the ISKNV MCP was confirmed by short read sequencing, and at the same location for all samples previously collected from Lake Volta outbreaks (*Alathari et al., 2023a*).

## DISCUSSION

This case study demonstrates the potential of using water samples for genomic surveillance of a large sized DNA virus, here for ISKNV infecting cultured tilapia in Ghana, using portable equipment in a farm setting. ISKNV was detected in both fish tissue samples and water samples collected from farm sites across Lake Volta and the in-field water sampling and sequencing method both distinguished between the different strains of the virus, and illustrated their relatedness. Sampling water within or close to the fish cages provides insight into the wider diversity of viruses on the farm than the more typical approach of tissue sampling because the latter is often based on a small number of fish, whereas the water may contain viral particles derived from many, potentially hundreds of fish, on the farm. Adopting the use of water sampling also avoids destructive sampling of fish with improved animal welfare benefits and reduced costs to the farmers.

ISKNV was detected in 81% of the fish sampled from the floating fish cages on Lake Volta in January 2023, with farm (F) having the highest viral load in both water and tissue samples of the farm sites studied. Although fish from farm (C) had very low concentrations of ISKNV in the body tissues sampled there was a relatively high concentration of viral particles in the surrounding water. Phylogenetic analysis of this water sample revealed it clustered with water samples collected from the upper region of Lake Volta. Farm (C) is surrounded by other tilapia farm cages in the Akuse region of Lake Volta and thus the likelihood is that ISKNV circulating strains may have been transported *via* the water from other nearby infected farms. In the current outbreak, some farmers reported a new trend of moribund tilapia, with fingerlings and juvenile fish being more susceptible than adult fish, and differing from that seen previously where no apparent age-related effect was reported (*pers. comms.*). In our analysis, ISKNV positive samples were seen at all maturation stages of tilapia, but the lowest concentration and infection rates were, in general, seen in adult fish. This may be as a consequence of ISKNV now being practically endemic in Lake Volta, and thus fish surviving to adulthood have likely been exposed previously and thus on re-infection with new outbreaks are able to mount a more effective immune response, thus limiting viral replication.

Farms (E) and (B) showed very low concentrations of ISKNV in the water, but there were relatively high titres of virus particles detected in the tissue samples. The floating fish

cages in both these two farms were located up to 12 km from the lake shoreline and from other farms, and this likely meant there was a far greater dilution of ISKNV from nearby fish in the water. Contrasting with this, water samples collected from farm (D) contained a high concentration of ISKNV, but for all of the fish, except one, sampled at this site there was a low body burden of the virus; this low viral template resulted in an inability to amplify it through our tiled PCR approach. We speculate this difference between the fish tissue and water titres of ISKNV might indicate a recent introduction of the virus to the farm and thus an early detection of virus presence through our water sampling approach. This highlights the potential utility of water sampling in monitoring for this pathogen, as previously applied in wastewater-based epidemiological studies for COVID-19 and Polio (*Gonzalez et al., 2020*; *Klapsa et al., 2022*). Another explanation could be that the fish have recovered from a viral episode at the time of sampling, with the surviving fish having overcome the infection. Further work is required to test these hypotheses.

Integrating this data set with our previously sequenced genomes collected from Lake Volta, phylogenetic analysis groups the majority of the 2023 sequences in a separate clade indicating that the ISKNV currently infecting tilapia in Lake Volta, are not a descendent of an ongoing /previous infection but rather an emergence of a different endemic strain, or a new introduction to the Lake; most likely through fish importation. Moreover, farm (D) clustered separately from all other samples, except for a tissue sample from farm (C), and revealed an additional mutation in ORF 117 (C105,539A) in both its tissue and water samples. ORF117 is a transmembrane protein (*Throngnumchai et al., 2021*) which plays a vital role in viral replication and virulence (*DiMaio, 2014*). The presence of this mutation likely explains why the water sample collected from farm (D) clustered separately from all other water samples. The close relatedness in the water and tissue sample in farm (D) highlights the capability of water sampling in detecting current, infective strains of ISKNV in fish. This was also confirmed when comparing the water and tissue samples of farm (F), where almost all SNPs were identical. It is also worth mentioning that the close relatedness of all but one of the samples collected in 2023, could indicate the same strain of ISKNV circulating in the water where this newly identified variant might be replacing the previous strain collected between 2018–2022. Sample (F.3.2) collected from farm (F) in 2023 clustered with samples collected in 2018 and might be a strain persisting from previous infections. Interestingly, water samples with less than 482 copies/µL of ISKNV, as calculated in our previous study (*Alathari et al., 2023a*), were able to recover more than 50% of the full ISKNV genome, yet this was not possible for tissue samples. This could be due to an increased diversity in the environmental samples, allowing for more primer binding to extracted DNA, or the tissue DNA from tissue samples may contain inhibitors that may affect the amplification (*Smith, 2021*).

The heat map of mutational frequencies highlighted the presence of different SNPs in some of the samples collected in 2023 when compared to samples collected from previous years. We observed the presence of four new mutations in samples collected from farm (E.2.1) and (F.4.1), which were lacking in all the tissue samples collected previously. At least one SNP was seen in the MCP, but samples from 2022 showed two SNPs in this location. The second SNP could have become a reversed mutation in samples collected in 2023, and

maybe have been corrected in the ISKNV genome due to its insufficient role in increasing the virus's fitness, or more simply the group that contains this second SNP was not sampled during this study. All samples collected from Ghana showed a mutation in ORF125 when compared to its reference genome. This ORF is an ankyrin repeat protein and also one of the major antigenic proteins and involved in modulating intracellular signalling networks during viral infections (*Guo et al., 2011*; *Throngnumchai et al., 2021*).

Short read sequencing of a water sample collected from farm (F) showed a SNP in ORF40 (C40,742T) and ORF58 (C51,475T). The SNP located in ORF58 had a variant frequency of 91.6%, with 8.2% showing the original reference sequence, indicating circulation of more than one variant in the farm. The SNP in ORF40 was not detected by the Geneious software, only by manual analysis. This mutation was present in both short and long read sequences, with the short read sequencing able to show at least two strains circulating the water. Both mutations were also seen in the water sample collected from farm (C) but not in farm(D), and could be the reason behind the (C) water sample clustering with (F) water sample. When comparing the water sample with the tissue samples, only two out of seven tissue samples collected from the same farm showed the same mutation at ORF058. This may indicate that the variant without the mutation at ORF058 derives from an historically earlier infection with a new mutation from a newly evolved variant. This is not presented in the heat map as the relevant samples, F.4.4, and F.4.3, generated a sequence recovery of less than 75% of the full ISKNV genome and were thus excluded from the analysis. Short read and long read sequencing produced a comparable number of SNPs and both approaches thus had the ability to detect the different variants.

In contrast to single-gene PCR approaches, whole genome sequencing can capture the full range of variants, providing vital information for vaccine and drug design. Other studies focusing on the MCP have shown their limitation in discriminating between viruses collected from different locations and at different time points (*Ayiku et al., 2023*). The portability of a next generation sequencer, and the invention of other portable technologies for amplicon generation and library preparation has led to long read sequencing being a preferred method for this analysis. These advancements have enabled performing studies like ours in remote and resource limited areas, with fast turnaround times, contrasting with that previously where the turnaround time at distant labs is in many months and likely unaffordable to many fish farm holders.

There are currently minimal disease control options for ISKNV and an urgent need for preventative measures. The approach we present in this article for Lake Volta, show that water sampling has great potential for use in identifying the ISKNV associated with infected fish, and for determining the variants circulating within the system and infecting the fish at the time of sampling. This could assist in improving disease prevalence estimates and in the detection of emerging variants. The fact that in many cases the water for the inland ponds for hatchery stages is drawn from the lake, is likely the reason for the presence(and repeated cycling) of ISKNV infections in all fish life stages. Seeking to combat this cycle of infections and re-infections of ISKNV, encouraging farmers to seek, and pressure for, farms designated free of ISKNV for their seeding stock would be a prudent step. Indeed, some larger farms with greater resources have already implemented this practice. Importantly,

this requires that the supporting systems for aquaculture programmes in Ghana need to enable disease free hatcheries to be established and this inevitably requires also training of fisheries officers and farmers in biosecurity practice and the associated resources to deliver this. In the future, we aim to optimise our field campaign by collecting samples throughout the year to account for factors such as fluctuation in water temperature, water quality, and sunlight exposure. This in turn will provide a better understanding of phylogeography and the epidemiology of ISKNV in Lake Volta over longitudinal gradients.

Currently, an epidemiological study for the management of ISKNV using the MinION sequencer remains costly for farmers in Ghana. While this technology promises scalable, accessible, and affordable sequencing, the cheapest device produced by ONT is the flongle, at the cost of $70, yet the cost of reagents used for library preparation is ~$600. Multiplexing is now feasible, using up to 96 barcodes, reducing the cost to approximately $7 per sample on the higher throughput MinION device used in this study. The methods applied here to ISKNV, in addition to its capability for application to reach remote regions, could be adapted for other viral infections affecting the growth and development of aquaculture. Combining field data with in-field genomic tools, supported by government programs, could provide opportunities to understand the genetic architecture of disease resistance, leading to new opportunities for disease control in real time. Finally, there are very few available whole genome sequences for ISKNV and other important fish viruses in the database, therefore, and routine sequencing of these viruses will benefit significantly, understanding of the mutations that occur across the genome, and their role in virulence and/or transmissibility of the viral diseases in aquaculture.

## ACKNOWLEDGEMENTS

We thank Janet Gyogluu Anchirinah, the Lake Volta local farmers and the Commission of Fisheries in Ghana involved in the investigation for their support and provision of information and samples. We also thank Joshua Quick for his insightful comments and Anke Lange, Victoria Jackson, Stephanie Andrews, Remy Chait, Exeter Microfluidics Facility, and Karen Moore for technical support and sequencing work.

### Funding

Shayma Alathari was funded on a PhD programme through the centre for Sustainable Aquaculture Futures, a joint partnership between the University of Exeter and the Centre for Environment, Fisheries and Aquaculture Sciences (Cefas). This project used equipment funded by the Wellcome Trust (Multi-User Equipment Grant award number 218247/Z/19/Z) to the Exeter Sequencing Centre. The funders had no role in study design, data collection and analysis, decision to publish, or preparation of the manuscript.

### Grant Disclosures

The following grant information was disclosed by the authors:

Sustainable Aquaculture Futures, a joint partnership between the University of Exeter and the Centre for Environment, Fisheries and Aquaculture Sciences (Cefas).
the Wellcome Trust (Multi-User Equipment: 218247/Z/19/Z.

## Competing Interests

The authors declare there are no competing interests.

## Author Contributions

- Shayma Alathari conceived and designed the experiments, performed the experiments, analyzed the data, prepared figures and/or tables, authored or reviewed drafts of the article, and approved the final draft.
- Andrew Joseph performed the experiments, authored or reviewed drafts of the article, enabled in-field sampling work and support in Ghana, and approved the final draft.
- Luis M. Bolaños analyzed the data, authored or reviewed drafts of the article, and approved the final draft.
- David J. Studholme analyzed the data, authored or reviewed drafts of the article, and approved the final draft.
- Aaron R. Jeffries analyzed the data, authored or reviewed drafts of the article, and approved the final draft.
- Patrick Appenteng performed the experiments, authored or reviewed drafts of the article, enabled in-field sampling work and support in Ghana, and approved the final draft.
- Kwaku A. Duodu performed the experiments, authored or reviewed drafts of the article, enabled in-field sampling work and support in Ghana, and approved the final draft.
- Eric B. Sawyerr performed the experiments, authored or reviewed drafts of the article, enabled in-field sampling work and support in Ghana, and approved the final draft.
- Richard Paley conceived and designed the experiments, authored or reviewed drafts of the article, and approved the final draft.
- Charles R. Tyler conceived and designed the experiments, authored or reviewed drafts of the article, and approved the final draft.
- Ben Temperton conceived and designed the experiments, authored or reviewed drafts of the article, and approved the final draft.

## Field Study Permissions

The following information was supplied relating to field study approvals (*i.e.*, approving body and any reference numbers):

Field experiments were approved by the Fisheries Commission, Ministry of Fisheries and Aquaculture Development, Accra, Ghana.

## Data Availability

The raw sequence reads are available at NCBI: PRJNA935699.

## Supplemental Information

Supplemental information for this article can be found online at http://dx.doi.org/10.7717/peerj.17605#supplemental-information.

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
