# Peer review of "In field use of water samples for genomic surveillance of infectious spleen and kidney necrosis virus (ISKNV) infecting tilapia fish in Lake Volta, Ghana"

_PeerJ, doi:10.7717/peerj.17605_

## Round 0.1 · original submission · Major Revisions

The paper focuses on a topic of interest and proposes methodologies that could be applied in areas other than those of this study. However, there are some major concerns regarding the methodology. It should be explained better than the viral outbreaks studied, those that are endemic to that area for this species, and the ones studied in the present work. Please also consider all the suggestions given by the reviewers in the revised version of the manuscript.

·

Basic reporting

Reviewer comments PeerJ-93974
This work presents a novel approach utilizing in-field epidemiological tools for genome sequencing surveillance of Infectious Spleen and Kidney Necrosis Virus (ISKNV) in tilapia farms. Specifically, the authors successfully demonstrated the detection of ISKNV in fish and water samples from Nile tilapia farms in Lake Volta, Ghana. Moreover, the utilization of water samples for virus detection represents a non-destructive sampling methods. By concentrating the virus genome from water samples surrounding infected fish cages, the authors effectively detected predominant ISKNV variants using whole-genome sequencing. The manuscript is well-structured, and the information provided is both clear and informative.
The findings of this study are significant as they highlight the feasibility of using water samples for monitoring of viral outbreaks, providing valuable insights into the dynamics of variant frequencies. The ability to track viral mutations and transmission chains in real-time is crucial for implementing timely control measures, such as isolating and quarantining infected farms, thus mitigating the spread of pathogens and minimizing economic losses. The methodology presented in this study has the potential to be adapted and implemented in other regions facing similar challenges, thus furthering our understanding of viral epidemiology in aquatic environments. Some comments include
- Regarding sample collection, it would be beneficial to consider collecting samples from different months to account for factors such as fluctuation in water temperature, quality, sunlight exposure, tidal movement, and fish density during various culture periods. Addressing these factors would enhance the robustness of the findings, particularly considering the potential impact on the persistence and survival of ISKNV in water. The author may try to address this in the discussion section.
Minor comments:
Line 63 Consider using a different term than “effectiveness of the virus”
Line 71 Give extra space “an epidemic (Luo,….)
Line 127 Insert an extra space “). The assembly
Line 166 forward and reverse primers
Line 177 For tissue samples,
Line 244 Why these three farms (C, D, F) have been selected.
Line 251, 263, 265 Check font format for consistency
Line 442-453 Consider adding additional relevant references or citations discussing “how the water sampling approach could be applied to monitor or provide early warning of virus outbreaks.
Line 474 “was not”
Line 475-476 Include additional references to support the hypothesis
Line 485 “was not”
Addressing the aforementioned comments and minor revisions would further enhance the clarity and impact of the study.

Experimental design

See the basic reporting

Validity of the findings

See the basic reporting

Additional comments

-

Reviewer 2 ·

Basic reporting

This manuscript aimed to demonstrate the utility of an in-field water sampling method for genomic surveillance of ISKNV in Lake Volta, Ghana. Using this in-situ detection approach, they identified ISKNV variants in water samples collected in the immediate vicinity of farms that generally corresponded to those found in the fish. The authors also compared data from previous field sampling to look out for changes in circulating strains. Generally, this is a useful study that is well-designed with a relevant research objective. It has some potential application in disease surveillance within the Ghanaian aquaculture sector and beyond.

Experimental design

There are couple of issues with the methodology that needs to be addressed.
In line 60, it is stated that ‘In Ghana, ISKNV, a Megalocytivirus, has become endemic in tilapia in Lake Volta’ following a series of outbreaks in 2018’. Are farms on the lake still experiencing outbreaks of the virus since its first detection in 2018? Since this study emphasizes on an ongoing outbreak of ISKNV (line 95), is there a new outbreak in the farms where samples were taken in January 2023, or was this across all farms on Lake Volta?
In line 97-98, what is a high density based-cage farm? Can authors provide the range of stocking density for these farms?
In line 111, details for the processing of tissue samples in the field must be provided.
Line 121, what is the excess liquid referring to? Is mentioned that the viral filters were preserved in RNALater® (line 105).
In line 147, the statement made is vague. Can the authors explain further how the positive control for water samples was used to test the efficiency of the DNA extraction method? Did they spike the water with the positive control strain and extracted DNA to see the recovery using the method described? Moreover, the description of ISKNV isolation using the BF-2 and or GF cell lines for DNA extraction is not very clear. The word ‘before filtration’ (line 150) is unnecessary and confusing. Generally, infected cells lines are fractionated and centrifuged to get rid of cell debris, and then the supernatant filtered to recover the virus for DNA extraction. This must be stated clearly. Also, only concentration of the extracted DNA is given (Supplementary Table 2) without the quality assessed. In lines 325-335, the genome coverage of ISKNV was related to the concentration of ISKNV in tissues and water samples. It may be prudent for authors to also consider the quality of DNA recovered from these two sample types (water vs. tissues) and looked out for possible inhibitors in the DNA extracts. The statement ‘unknown mechanisms’ is far- fetched.

Validity of the findings

Lines 431-433 mentioned transmission of the virus in water. Is there enough evidence to support this? Were attempts made to detect ISKNV in water samples taken some distance away from the cages?
Line 436 add a reference after ‘reported.’
Lines 450-453; No convincing evidence presented to support both hypotheses. What made the fish possibly recover from earlier infection? What treatment was imposed? Some evidence points to the fact that heat shock therapy, which is one of the most adapted interventions against ISKNV by farmers on Lake Volta does not get rid of the virus from infected fish (https://doi.org/10.1016/j.aquaculture.2023.740330).
Lines 541 mentioned affordability, but this manuscript did not assess the cost for in-field water sampling, processing and whole genome sequencing of the virus. Have the authors considered the cost of reagents and logistics needed for the analysis and maintenance of the portable equipment, especially in remote and resource limited areas in Africa as mentioned? Some cost-benefit analysis with existing methods may be required.

Additional comments

Table 1 has farm locations. These are not regions, but town within specific districts in Ghana. This should be checked and corrected.

There are also a couple of errors -repetition of sentences/abbreviations without full names that need to be checked (lines 163m 197, 201, 403- 407, 417 etc.)

---

## Round 0.2 · accepted · Accept

I am pleased to confirm that your paper has been accepted for publication in PeerJ.

Thank you for submitting your work to this journal.

Reviewer 2 ·

Basic reporting

The manuscript is significantly improved. I have no further comments.

Experimental design

Good

Validity of the findings

Good

Additional comments

No further comments